# Aging of the Arterial System

**DOI:** 10.3390/ijms24086910

**Published:** 2023-04-07

**Authors:** Roberto Castelli, Antonio Gidaro, Gavino Casu, Pierluigi Merella, Nicia I. Profili, Mattia Donadoni, Margherita Maioli, Alessandro P. Delitala

**Affiliations:** 1Department of Medicine, Surgery and Pharmacy, University of Sassari, 07100 Sassari, Italy; 2Department of Biomedical and Clinical Sciences Luigi Sacco, Luigi Sacco Hospital, University of Milan, 20157 Milan, Italy; 3Cardiology Unit, Azienda Ospedaliero, Universitaria di Sassari, 07100 Sassari, Italy; 4Department of Biochemical Science, University of Sassari, 07100 Sassari, Italy

**Keywords:** arterial stiffness, aging, vascular aging

## Abstract

Aging of the vascular system is associated with deep changes of the structural proprieties of the arterial wall. Arterial hypertension, diabetes mellitus, and chronic kidney disease are the major determinants for the loss of elasticity and reduced compliance of vascular wall. Arterial stiffness is a key parameter for assessing the elasticity of the arterial wall and can be easily evaluated with non-invasive methods, such as pulse wave velocity. Early assessment of vessel stiffness is critical because its alteration can precede clinical manifestation of cardiovascular disease. Although there is no specific pharmacological target for arterial stiffness, the treatment of its risk factors helps to improve the elasticity of the arterial wall.

## 1. Introduction

Cardiovascular disease is the leading cause of death worldwide. Several risk factors have been identified and, among these, increased arterial stiffness has gained attention due to its correlation with many cardiovascular endpoints [1]. Studies reported the determinants of arterial stiffness [2], which contributes to the stiffening of the wall artery. The structure of the vessel wall varies along the vascular tree: the media layer of small arteries has small amounts of elastin and elastic lamellae as compared to central vessels, which have higher concentrations of vascular muscle cells. Thus, the greater elasticity of large arteries allows to dampen the pulsatility of ventricular ejection and to convert into a relative steady flow blood downstream at the site of arterioles. This mechanism protects organs from pulsatile energy [3].

The propagative model based on the visco-elastic hypothesis is one of the most reliable models used to describe and understand the hemodynamic of the circulatory system. Here, the elasticity of the tube generates a forward pressure wave that travels along the tube. However, the intrinsic characteristics of the tube (branch points and high resistance) generate retrograde wave reflection. This mechanism is critical for the diastole phase and promotes coronary perfusion.

During systole, the stroke volume is forwarded directly to the peripheral tissues, but part of it involves the stretching of the aorta and central arteries walls. Thus, part of the energy is stored in the vessel walls, which, during diastole, recoils the aorta, squeezes the accumulated blood forward into the peripheral tissues, and ensures a continuous flow. This ability mainly depends on the stiffness of the artery, whose loss impairs the protective mechanisms and somewhat facilitates damage of the microvascularization of organs, in particular brain and kidney [3]. Indeed, stiff arteries need high pressure to be stretched, and this loss of energy causes an intermittent flow and pressure and, in addition, an excessive flow at the distal artery level.

In this review, we summarize the pathophysiological events that contribute to the development of arterial stiffness (Figure 1) strategies aimed to reduce the stiffening of the arterial wall. We particularly focused on:

## 2. Arterial Wall Remodeling during Aging

### 2.1. Collagen and Elastin

The ratio between collagen and elastin in the vascular wall ensures the optimal arterial compliance and is guaranteed by a balance between the production and degradation processes. Collagen limits the stiffness due to its stiff properties, whereas elastin provides the flexibility and extensibility of the artery wall.

Vascular aging is characterized by deep changes in arterial structures, with specific regional differences. In the central artery, collagen fibers increase [4], whereas the number of elastic fibers and smooth cells decreases. In particular, type I was found to be the most prevalent type of collagen before and after 50 years of age both in the arch and lower abdominal aorta. On the other hand, type III collagen decreases from the arch to abdominal aorta (from 24% to 18%) [4]. Smooth cells in the tunica media, which are responsible for synthesizing elastin, decrease with aging and, therefore, elastic fibers markedly decrease. Additional factor that concours to the reduction of elastic fibers is the increase of elastase activity in the abdominal aorta and the reduction of tropoelastin expression. This molecule, which is secreted by elastogenic cells, is the soluble precursor of elastin, and its expression is decreased by 50% each decade [5]. These processes are strictly connected: the fracturing of elastin fibers at the tunica media level causes a collagenous remodeling. Indeed, other than an increase in proteoglycans, collagen becomes thicker and more linear [5]. A key role of the vascular remodeling process is played by matrix metalloproteinases, which are enzymes capable of hydrolyzing components of the extracellular matrix. The action of these enzymes is regulated by different factors whose alteration results in the overproduction of collagen and a reduction of functional elastin [6]. Animal studies allowed the evaluation of the composition of different segments of the aorta artery. The thoracic aorta shows a steady collagen density during aging, whereas the latter was found to be increased in the abdominal aorta. On the other hand, elastin decreases with aging in both the thoracic and abdominal aorta [7]. Studies on humans are rather scarce and somewhat controversial. Indeed, some reports showed a decrease in collagen [8], whereas others found an increase of collagen during aging [9,10]. Additional studies reported no changes [11].

Peripheral arteries, compared to central arteries, are more muscular and stiffer. However, they are less prone to vascular aging. For example, the stiffness of the femoral artery does not change with aging in men or women, but a marked increase of its stiffness was documented in women over 60 years of age [12]. The carotid artery is less rigid and more distensible than the femoral artery and has a smaller diameter [13]. The distensibility of the carotid artery and cross-sectional compliance decrease linearly with aging [13]. Similar to the femoral artery, the stiffness of the carotid artery is higher than the aorta prior to the 50 year mark, but showed a less sharp increase in stiffness with aging [14]. Adiposity seems to play an important role in this context. Indeed, a study from the Baltimore Longitudinal Study of Aging reported that abdominal adiposity is associated with carotid femoral stiffness, which is explained in part by leptin but not adiponectin [15].

### 2.2. Calcification

A clear factor of reduced wall distensibility of the arterial wall during aging is the calcium deposition within the arterial wall [16]. Although it is not clear why aging is characterized by an increased calcification of the vessel wall, it has been postulated that inflammation and oxidative stress may play a role, as both occur during aging. Indeed, they stimulate vascular calcification and decrease mitophagy and autophagy [17]. The severity of calcium deposition varies according to the type of artery involved, and the presence of comorbidities can facilitate the calcification process [18]. Gender-effect has been reported by one study that found specific cardiovascular risk profiles for different arteries. Indeed, calcification at extracranial and intracranial carotid arteries is more severe in women with arterial hypertension, whereas in men, the strongest association was found between arterial hypertension and vertebrobasilar arteries [19]. Another study from Rotterdam survey also showed that chronic obstructive pulmonary disease plays a clear central role as a risk factor for the presence of calcification at different sites (coronary, aortic arch, extra- and intra-cranial carotid artery) [20].

### 2.3. Endothelial Dysfunction and Intima-Media Thickening

Vascular tone is a process of alternating vasoconstriction and vasodilation. Nitric oxide regulates vasodilation, and any alteration of its production leads to endothelial dysfunction. This, in turn, causes an increase of oxidative stress, damages vessels, and promotes arterial stiffening [21]. Recent evidence also suggests that autophagy is reduced in the endothelium of aging people and may play a role in the development of endothelial dysfunction [22].

The thickening of the intima-media tunica layer is a marker of subclinical atherosclerosis. Large arteries are characterized by a 15–40% increase of intima-media thickness. Tunica media at the aorta level does not substantially change with aging, whereas tunica intima shows a thickening that is contributing factor for the increase in the aortic wall diameter. Atherosclerosis is characterized by the presence of vascular injury, which causes the migration of smooth muscle cells from the tunica to the intima. The presence of a thickened intima tunica causes a reduction in its elasticity. Intima media thickness at the carotid level shows an age-specific pattern of increased rates: from 0.02 mm/decade at the age of 50 to 0.05 mm/decade at age of 80 years. Furthermore, the rates are higher in men than in women [23]. However, a recent meta-analysis of cohort studies reported that although intima media thickness at the carotid level is associated with cardiovascular disease, its change is not related to an increased risk of a future event [24].

### 2.4. Genetic Determinants

Studies focused on the heritability of arterial stiffness found that genes explain 23–50% of the variability in arterial stiffness and act independently of traditional cardiovascular risk factors. However, for most of them, the underlying mechanism has not been fully elucidated at the molecular level.

*CUL3* gene codes for cullin-3, which is a protein that belongs to the cullins family and is a component of the cullin-RING E3 ubiquitin ligases complex [25]. Mutations of the *CUL3* gene can cause pseudohypoaldosteronism type 2, which is characterized by hypertension, hyperkalemia, and increased arterial stiffness. The reduction of the degradation of cullin-3 substrate RhoA, secondary to mutations of cullin-3, causes an increased Rhoa/ROCK signaling. An additional proposed mechanism is the reduced expression of subunits of soluble guanylyl cyclase (sGCα1 and sGCβ1), which leads to the impaired production and activity of cGMP [25]. A recent GWAS found variants at four loci correlated with arterial stiffness index [26]: *TEX41*, *FOXO1*, *MRVI1*, and *C1orf21*. The first locus has also been associated with coronary artery disease [27]. The *FOXO1* gene, located within the O class of the forkhead family of transcription factors, regulates mindin, which has a role in vascular smooth cell proliferation. The *MRVI1* locus encodes the MRVI1/IRAG protein that is involved in smooth muscle contractility. The function of *C1orf21* is not currently known. Another study reported an association adrenomedullin gene expression and indexes of arterial stiffness (reflection index and stiffness index) [28]. Adrenomedullin is a circulating vasoactive protein, and the reported correlation might explain its role in the modulation of vascular endothelium. Finally, a meta-analysis of genome-wide association data of nine community-based cohorts reported a common genetic variation in a locus in the *BCL11B* gene [29], which has already been associated to left ventricular hypertrophy [30]. The *BCL11B* gene codes for a promoter transcription factor (COUP-TF) that modulates the angiopoietin-1 and vascular endothelial growth factor pathway. *BCL11B* may also regulate T-cell production because its specific deletion causes an increase in proinflammatory T-cells that could potentially promote inflammation, fibrosis, and stiffening [31].

It should be noted that the association between genetic polymorphisms and arterial stiffness is not univocal, and several reason can be found. The most important source of inconsistent results is the different methods used to assess the arterial stiffness. Regional differences (central or peripheral arteries) and selection of the study sample can be considered as additional confounding factors.

## 3. Method to Assess Arterial Stiffness

Pulse wave velocity (PWV) is a surrogate of arterial stiffness and represents the velocity at which the blood pressure pulse moves down the vessels. The Bramwell-Hill equation can theoretically explain the PWV, providing the relationship between arterial distensibility and PWV. In clinical practice, PWV is calculated as the ratio between distance between two measuring sites and the ratio of time taken by the pulse to travel between the measuring sites. Several methods have been created to assess PWV.

Carotid-femoral PWV (cf-PWV) is the most widely used method to assess the arterial stiffness. Signals are collected from the carotid and femoral artery, which are both close to the aorta, thought different commercially available systems. The collection of data can be done simultaneously or sequentially guided by ECG-gated. The appropriateness of its measurement relies on the assessment of the carotid-femoral path length, which can be simply calculated using 0.8 times the distance between the two measuring sites [32]. Although cf-PWV is the most reliable method used to evaluate arterial stiffness, it should be noted that it does not assess the ascending aorta and part of the aortic arch.

Magnetic Resonance Imaging (MRI) is probably the only method that can provide accurate anatomical imaging and transit time and, in addition, allows the assessment of the segments of aorta not evaluated by the cf-PWV [33]. However, costs, timing, and local availability do not allow its use in the general population.

Brachial-ankle PWV (ba-PWV) is an alternative method to assess PWV that is frequently used in Asia [34]. ba-PWV measures the transit of pulse wave from the brachial and ankle artery. The distance between the two measuring sites is calculated using the linear regression of body weight. However, the large distance between the two arteries and the different structure of arteries found along the journey leads to ambiguity with regard to its interpretation.

The cardio-ankle vascular index (CAVI) is derived from the arterial stiffness index β and was introduced by Japanese researcher to obtain pressure-independent values of arterial stiffness. Although it displays a good reproducibility among different vascular diseases [35], there are a number of clinical conditions that limit its use, such as aortic stenosis, atrial fibrillation, and peripheral arterial disease [36]. In addition, CAVI assesses both elastic and muscular arteries, and, therefore, analyzes a heterogeneous phenotyping.

Pulse wave can be also assessed indirectly through an analysis of wave reflection, which is quantified by augmentation index (Aix) at the carotid or at radial artery. Aix is determined by the amplitude of the reflection wave, by the distance to the reflected site, and by the cardiac cycle. Due to its intrinsic limits, Aix also needs additional parameters (pulse pressure and systolic blood pressure) and is often assessed together with the measurement of PWV.

## 4. Clinical Risk Factor for Arterial Stiffness

Stiffening of the arterial wall is a physiological process strictly connected with aging [37]. Several risks factors have been documented so far [38,39,40].

### 4.1. Hypertension

Arterial hypertension has a high frequency in the general population and is deeply connected with arterial stiffness (Table 1). Although the pathophysiology is not clear, the relation between arterial stiffening and hypertension has different causes. The mineralcorticoid receptor is expressed in endothelial cells, myocytes, and other specific cells, where it directly regulates the transcription of vascular gene and vascular smooth cells muscle [41]. The expression of the mineralcorticoid receptor increases with aging, and its activation contributes to cardiovascular disease. Indeed, it may promote cardiac fibrosis, hypertrophy, and arterial stiffening [42]. Specific treatment with mineralcorticoid receptor antagonists seems to improve these effects, as demonstrated by the reduction in specific circulating biomarkers [41]. Another study found that transforming growth factor β1 (TGF-β1) has a key role in cardiovascular disease. Indeed, Podzolkov et al., reported an increased concentration of TGF-β1 in patients with uncontrolled arterial hypertension and further demonstrated a positive association with arterial stiffness [43]. Rho/Rho-associated coiled-coil containing kinases 1 and 2 (ROCK1 and ROCK2) modulates TGF-β1 and regulates cellular contraction and vascular remodeling. In particular, ROCK1 and ROCK2 mediate age-related aortic stiffening in mice and can, therefore, be a therapeutic target for preventing increased arterial stiffness [44].

Previous reports suggest that increased pulse pressure accelerates elastin degradation through an increase in pulsatile wall stress [45]. This means that hypertension accelerates arterial wall aging, thus increasing the stiffness. However, other studies clearly show that the presence of increased levels of arterial stiffness are associated with an increased risk of incident hypertension in normotensive patients [46]. Although the exact cause–effect relationship still needs to be elucidated, different hypotheses can be made. Hypertension may cause increased arterial stiffness by causing vascular damage and elastin rupture. Conversely, arterial stiffness widens pulse pressure, which is connected to increased systolic blood pressure. Several lines of evidence suggest that arterial stiffness linearly increased in both normotensive and hypertensive subjects [47], with a similar slope of curve. A central role is played by systolic blood pressure. Indeed, it has been demonstrated that in older patients, systolic blood pressure increased along with arterial stiffness, independently from diastolic blood pressure [48]. Longitudinal studies allowed to create specific trajectories of blood pressure in young individuals after 30 years of follow-up [49]. Authors demonstrated that baPWV was associated with systolic and diastolic blood pressure in subjects that develop a rapid increase in blood pressure, but <125 mmHg, and in those who experienced the highest increase in blood pressure.

### 4.2. Diabetes Mellitus

The effect of inappropriate glycemic control on arterial stiffness is well acknowledged and is multifactorial. Advanced glycation end products (AGE) are highly oxidizing products of nonenzymatic reactions between reducing sugar and biological proteins (e.g., collagen) or lipids [50]. They mainly derive from endogenous production but are also abundant in specific food. AGE interact with specific receptors (RAGE) that are expressed in smooth muscle cells, endothelial cells, and epithelial cells, and cause the cross-linking of collagen fibers. This, in turn, causes an increase in collagen in the arterial wall due to an increased resistance to its enzymatic proteolysis [51]. The soluble isoform of RAGE (sRAGE) circulates in the blood and binds to AGE to avoid the interaction between AGE and RAGE, thus protecting the endothelium from the above-mentioned effects. The importance of sRAGE has been documented in animal and human studies. Indeed, mice treated with sRAGE suppressed accelerated diabetic atherosclerosis [52]. In humans, patients with reduced levels of sRAGE had an increased risk of coronaropathy and atherosclerosis, independently of diabetes [51]. Therefore, the effect of AGE/RAGE on arterial stiffness is clearly linked. Indeed, skin AGE significantly and positively correlates with aortic stiffness [53]. In addition, it has also been demonstrated that reduced levels of circulating sRAGE independently increase the risk of aortic stiffness [54]. Clinical consequences of the detrimental effect of diabetes on arterial stiffening were reported by Markus et al., who analyzed over 1000 individuals from two independent cohort studies [55]. The authors demonstrated that a 1 mmol/L higher fasting glucose was associated with a 0.12 mmHg/m^2.7^/mL greater arterial stiffness index.

### 4.3. Atherosclerosis

The association between atherosclerosis and arterial stiffness is not clear. Although many studies reported that aorta stiffening increases with plaque burden [56], it should be noted that both conditions are commonly found during aging, and increased arterial stiffness can be found in the absence of atherosclerosis [57]. Hypercholesterolemia is obviously the key factor implicated in the stiffening of atherosclerotic arteries. It has been suggested that the increased arterial stiffness found in hypercholesterolemic patients has three major causes: firstly, dyslipidemic patients have increased level of microparticle release and reduced number of endothelial progenitors [58]. In addition, calcification within the atherosclerotic plaque may play a major role, as demonstrated by Cecelja et al., who reported that arterial stiffness was mostly related to the propensity of plaques to calcify [59]. Finally, a role of oxidative stress was proposed by Zagura et al., who reported that arterial stiffness with osteopontin and oxidized low density lipoprotein are both involved in the atherosclerotic process [60]. Clinical evidence from survey studies showed that arterial stiffness, assessed as caPWV and at the common carotid artery, was strongly associated with atherosclerosis at different sites (carotid artery and aorta) [61].

### 4.4. Chronic Kidney Disease

The cross-talk between the kidney and heart has been known about for decades, and the functional abnormalities of one reflect on the other. The reduction in aortic stiffness causes a loss of buffering of systolic volume, which could damage the kidney vasculature and, therefore, its function. Patients with chronic kidney disease are prone to developing vascular calcification [62]. Calcium tended to deposit in the tunica media, and the reduction in renal phosphate excretion has a central role in this context. Indeed, hyperphosphatemia promotes the activation of Toll-like receptor four and NK-Kappa B in vascular smooth muscle cells, both of which are involved in the development of atherosclerotic lesions [63]. Additional effect of hyperphosphatemia (activation of pro-inflammatory molecules, increased reactive oxygen species production) also caused mitochondrial dysfunction. The reduction in uric acid excretion has an additional role in the development of the stiffening of arteries. Indeed, it decreases nitric oxide synthetase and the proliferation of vascular smooth muscle cells [64]. Furthermore, it increases the production of angiotensin II, which contributes to arterial stiffness. Chronic kidney disease also causes a progressive accumulation of AGE due to an increase in their production and a reduction in their elimination. The role of AGE on arterial stiffness has been discussed above. Finally, a role of endothelin 1 in both renal and cardiovascular disease has been clearly documented. Other than its potent vasoconstrictor activity, endothelin 1 acts on specific receptors (ETa and ETb) and causes endothelial dysfunction, calcification, inflammation, and vasoconstriction [65].

Chronic kidney disease may also impair collagen cross-linking and matrix remodeling. In this context, lysyl oxidase has recently been proposed by Sharma et al., as a cofactor for the development of increased arterial stiffness in mice with early impaired kidney function [66]. Authors demonstrated that mice treated with lysyl oxidase inhibitors showed a decrease in cross-linked collagens and PWV compared to vehicle treatment. Studies on humans reported clinical consequence of arterial stiffness on chronic kidney disease. Findings from Rotterdam Study showed that an increase in estimated glomerular filtration rate was associated with lower PWV. Authors also demonstrated that mean PWV values increased along with the quartile of glomerular filtration rate [67]

## 5. Behavior Risk Factors

Arterial stiffness is associated with different behavior risk factors, some of which are strongly related, whereas for others, the association is less clear due to issues correlated to their quantification.

### 5.1. Smoking

Smoke is one of the most important risk factors for cardiovascular disease. Studies showed that smoking is a strong risk factor for increased arterial stiffness [68] and directly associated with the number of daily cigarettes smoked [69]. The detrimental effect of smoking is greater when combined with the other risk factors, such as dyslipidemia, inflammation, and arterial calcification. In addition, the damage induced by smoke can be partially reversed only after 10 years of smoking cessation [69]. Subjects exposed to passive smoking show similar alterations [70].

### 5.2. Alcohol

The effect of alcohol on arterial stiffness is complex. Indeed, it has been demonstrated that a chronic, low dose of alcohol can decrease arterial stiffness [71] and probably decrease the oxidative stress, promote an increase of HDL, and reduce insulin resistance. This effect is age independent, and it has been also demonstrated that a small amount of alcohol can also acutely decrease PWV by increasing nitric oxide [72]. On the other hand, moderate and heavy drinkers have increased arterial stiffness, regardless of age, gender, and menopausal status [71,73].

### 5.3. Physical Activity

The beneficial effect of physical activity is well acknowledged and not limited to arterial stiffness. Studies reported that physical activity can lower PWV in some cases and reduce the physiological age-related increase in others [74]. Regular training (more than 20 min of aerobic exercise at least 3 times per week) improves arterial stiffness [75], probably thanks to changes in arterial wall stress and vasodilation mediated by nitric oxide.

## 6. Increased Arterial Stiffness and Clinical Outcome

### 6.1. Atrial Fibrillation

Atrial fibrillation is one of the most prevalent arrhythmias in the general population, and its incidence dramatically increases with aging [76]. Several studies reported that elevated arterial stiffness increases the risk of atrial fibrillation [77]. Although the exact pathophysiology is still not known, a central role of pulse pressure has been hypothesized. Indeed, elevated pulse pressure, which contributes to increased arterial stiffness, could lead to left atrium distension, thus contributing to the development of atrial fibrillation. Studies showed that patients with atrial fibrillation, without any other cardiovascular risk factors, have higher arterial stiffness, this suggesting that modification of the arterial wall artery could be implicated in the onset of this arrhythmia [78]. Analysis from the three population-based cohort studies confirmed that greater arterial stiffness, together with higher intima-media thickness and the presence of plaque at the carotid level, are associated with a higher incidence of atrial fibrillation [79].

### 6.2. Stroke

Stroke is the second leading cause of death and is correlated to loss of ability. Studies found that increased arterial stiffness is associated with the onset of acute stroke, even fatal [80], independently from other cardiovascular risks factors, such as arterial hypertension and dyslipidemia. Interestingly, cf-PWV assessed 1 week after an acute stroke was associated with a poorer functional outcome [81]. The exact mechanism beyond this association is still not clear, but the recent evidence that cerebral small vessel disease is linked to arterial stiffness may suggest that vascular stiffness and acute stroke could share the same pathophysiological mechanism [82]. A recent individual participant data meta-analysis strengthened this concept. Indeed, it included over 22,000 subjects and showed that carotid stiffness is associated with an incident of stroke (HR 1.18%) [83]. Interestingly, the association was independent from cardiovascular risk factors and aortic stiffness.

### 6.3. Declined Cognitive Function

Cognitive decline has become a major health issue due to the increased life expectancy around the world. Several risk factors for its development have been identified so far, and arterial stiffness has gained a major role as a contributing factor to dementia [84]. The association between arterial stiffness and cognitive impairment is not fully elucidated, but central hemodynamics play a pivotal role. Indeed, increased arterial stiffness causes an excessive pulsatile energy in the brain microvascular bed due to insufficient flow wave damping, thus promoting the onset of microbleeds [85]. Studies on murine models also showed that the short-term pharmacological induction of arterial stiffness does not cause additional effects on mouse models of Alzheimer, suggesting a pathogenetic role of long-lasting arterial stiffness [86]. Recent studies also reported a positive association between local amyloid-β and regional tau burden and aortic stiffness [84], but its exact cause–effect relation deserves further studies. Indeed, some studies failed to identify arterial stiffness as an independent risk factor for cognitive decline [87].

## 7. Pharmacological Treatment of Arterial Stiffness

Several studies have been published aiming to test which therapy is the best option to treat arterial stiffness. However, no specific drug has been synthetized to specifically modulate the arterial stiffness, and almost all previous studies reported the effect on arterial stiffness during the treatment of other specific diseases (Table 2). This means that studies in literature reported the effect on the relevant disease, together with the effect on arterial stiffness. Thus, it is not possible to clearly discern a direct effect of the drug on arterial stiffness, or rather an indirect effect secondary to the treatment of another disease.

### 7.1. Antihypertensive Drug

The different classes of drug used to treat arterial hypertension showed different effects of arterial stiffness. Among these, angiotensin receptor blockers and ACE inhibitors showed the most prominent effect in reducing stiffness [88]. Other than a direct effect of reducing blood pressure, which is common to all agents used to treat hypertension, drugs that antagonize the renin-angiotensin system may further act as antifibrotic agents. A recent meta-analysis strengthened this concept [89]. In addition, the modulation of stiffening seems to also be related by the dose of ACE inhibitors. Indeed, Tropeano et al., reported that the reduction in arterial stiffness in a population of hypertensive diabetic patients treated with Perindopril 8 mg was superior to the lower dose (4 mg) [90]. β-blockers were also effective in indirectly lowering the stiffness by reducing the heart rate, with a clear impact on the visco-elastic properties of the arterial wall. The effect of the other hypertensive drugs on arterial stiffness is less evident or even absent.

### 7.2. Antidiabetic-Drugs

Diabetes mellitus is an independent risk factor of arterial stiffness, and its treatment intrinsically modulates the stiffening in the artery wall by reducing AGE and its interaction with RAGE. Among the drugs used to treat diabetes, glitazones are the most studied and can decrease aortic stiffness [91]. Araki et al., also demonstrated that metformin may improve arterial stiffness by increasing adiponectin [92]. Selective sodium-glucose cotransporter inhibitor (SGLT2i) is a novel class of antidiabetic drugs that showed different effects beyond the glycemic control. A recent prospective study reported a decrease in the arterial stiffness in 32 patients treated with dapaglizofin for 12 months [93]. Similar results have been reported by Hong et al., in 140 diabetic patients treated with dapaglizofin for 6 months [94]. Interestingly, the use of gliflozins is associated with a 11% reduction in vascular stiffness, which was independent from the use of glitazones [95].

### 7.3. Lipid-Lowering Drugs

Statins are commonly used to treat dyslipidemia and act by inhibiting HMG-CoA reductase, a key enzyme of the mevalonate pathway. Beyond the direct effect of reducing cholesterol, statins could improve arterial stiffness. Indeed, although some previous studies reported an improvement of aortic as well as carotid stiffness, others failed [96,97,98]. Though this inconsistency may be due to methodological issues, it should be noted that statins may play a role in reducing arterial stiffness by modulating inflammation. However, at the same time, it seems clinically not significant. The new lipid-lowering drug, proprotein convertase subtilisin/kexin type 9 (PCSK9) inhibitors, plays an important role in the cholesterol metabolism and targets the degradation of low-density lipoprotein receptors [99]. PCSK9 inhibitors beneficially affect all risk factors of arterial stiffness, and a recent study showed a reduction of PWV in familial hypercholesterolemia subjects without atherosclerotic cardiovascular disease [100]. Similar findings have been reported in subjects treated with ezetimibe in combination with statins [101].

### 7.4. Perspective

Many researchers are currently testing different treatment approaches to improve age-related vascular dysfunction in humans. Among these, Murray et al., tested the effect of mitochondrial-targeted antioxidant supplementation on 45 healthy older men and women aged at least 60 years. The study, which was a randomized, placebo-controlled, double blind, phase IIa clinical trial, showed that its use in older adults without clinical disease improves endothelial function and reduces arterial stiffness [102]. Similar effects have been documented by another study [103], which reported that the improvement of brachial artery flow-mediated dilation with an improvement in endothelium-specific nitric oxide-mediated dilation. Acute mitochondrial antioxidant intake also improves exercise tolerance [104].

**Table 2 ijms-24-06910-t002:** Effect of drugs on pulse wave velocity.

Drug Class	PWV	Effect
Anti-hypertensive drugs		
ACE Inhibitors	↓	↓ RAS and acts as antifibrotic agents [105]
Angiotensin receptors blockers	↓	↓ RAS and acts as antifibrotic agents [106]
Calcium channel antagonists	↔ or ↓	None or reduce wave reflection [107]
β-blockers	↓	Reduce heart rate and modulate visco-elastic properties of arterial wall [89]
Nitrates	↔	None documented
Diuretics	↔	None documented
Aldosterone antagonists	↔ or ↓	None or modulate fibronectin expression and vascular tone [108]
α-blockers	↔ or ↓	None or increase nitrogen oxygen [109]
Antidiabetic drugs		
Glitazones	↓	↓ AGE and its interaction with RAGE [91]
Metformin	↔ or ↓	↓ AGE and its interaction with RAGE; increase adiponectin [110]
Lipid-lowering drugs		
HMG-CoA reductase inhibitors	↓	Modulate inflammation [97]
PCSK9 inhibitors	↓	Modulate inflammation [111]
Ezetimibe	↓	Modulate inflammation [101]

Abbreviations: RAS, renin-angiotensin system; HMG-CoA, β-Hydroxy β-methylglutaryl-CoA; PCSK9, Proprotein Convertase Subtilisin/Kexin type 9; ↑ = increase; ↓ = decrease.

## Figures and Tables

**Figure 1 ijms-24-06910-f001:**
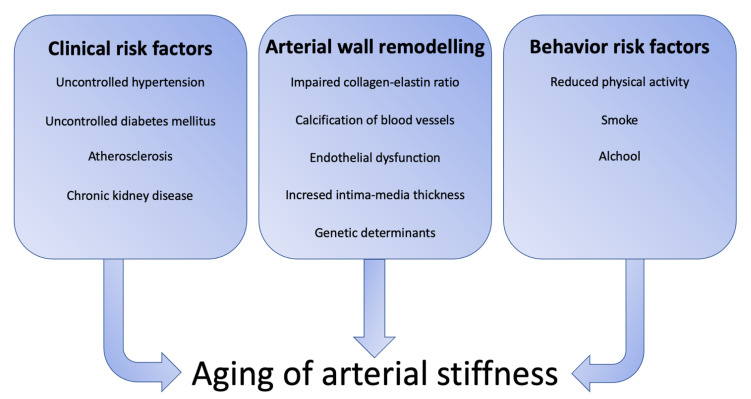
Summary of factors that contribute to the development of arterial stiffness.

**Table 1 ijms-24-06910-t001:** Effects of clinical risk factors on arterial stiffness.

Disease	Molecule/Clinical Parameter	Effects
Arterial hypertension	↑ Pulse pressure ↑ Systolic blood pressure	Elastin rupture Direct vascular damage
Diabetes mellitus	↑ AGE/RAGE ↓ AGE/sRAGE	↑ resistance to enzymatic proteolysis of collagen ↑ Collagen in arterial wall
Chronic kidney disease	↑ Phosphoremia	Activation of Toll-like receptor four and NK-Kappa B Activation of pro-inflammatory molecules ↑ reactive oxygen species production
	↑ Uric acid	↑ nitric oxide synthetase ↑ proliferation of vascular smooth muscle cells ↑ production of angiotensin II
	↑ AGE/RAGE	↑ resistance to enzymatic proteolysis of collagen ↑ Collagen in arterial wall
	↑ Endothelin 1	Endothelial dysfunction Calcification Inflammation Vasoconstriction

Abbreviations: AGE, Advanced glycation end products; RAGE, receptor of AGE; sRAGE, soluble isoform of RAGE; ↑ = increase; ↓ = decrease.

## Data Availability

Not applicable.

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
