# Peer review of "Aging of the Arterial System"

_ijms, 2023, doi:10.3390/ijms24086910_

Round 1
Reviewer 1 Report
Review: Aging of arterial stiffness
The author has beautifully put together review literature involved in the aging of arterial stiffness. It encompasses the genetic, clinical and behavioral association of the cardiovascular disorder with arterial stiffness. At the end the authors have shed the light on the pharmacological treatment for arterial stiffness. The review looks good to me and would like to ask the author to make these small changes.
Major:
-
The author can increase the part of the Genetic determinants of arterial stiffness by adding the genetic epidemiological studies performed for arterial stiffness.
Minor:
-
Line 27 Abstract, there is no need to change the line.
-
Line 33 Please put the reference after “many cardiovascular endpoints”.
-
Line 34 no need of “variously” or change to better adjective.
-
Line 45 changes “allows” to “involves”.
-
Line 96 instead of “in turns” changes to “in turn”.
-
Line 97-98 "evidence also suggests" should be "evidence also suggests".
-
Line 112,118,119, 120 please italicize the gene name.
-
Line 146 Please write the full of “ASIA”.
-
Line 157, please change “carotid o radial artery” to “carotid radial artery”
-
In Table 1 the author for AGE/RAGE has written See above. Please be specific or write some bullet points.
Author Response
Reply to Reviewer 1
- “The author has beautifully put together review literature involved in the aging of arterial stiffness. It encompasses the genetic, clinical and behavioral association of the cardiovascular disorder with arterial stiffness. At the end the authors have shed the light on the pharmacological treatment for arterial stiffness. The review looks good to me and would like to ask the author to make these small changes.”
Comment: We wish to thank the Reviewer for appreciating our work.
- “The author can increase the part of the Genetic determinants of arterial stiffness by adding the genetic epidemiological studies performed for arterial stiffness.”
- Comment: We agree with Reviewer’s comment. We increased the section “Genetic determinants” adding specific studies focused on arterial stiffness.
- “Line 27 Abstract, there is no need to change the line.”
Comment: We removed the paragraph.
- “Line 33 Please put the reference after “many cardiovascular endpoints”.”
Comment: we added the appropriate reference.
- “Line 34 no need of “variously” or change to better adjective.”
Comment: we deleted the term “variously”
- “Line 45 changes “allows” to “involves”.”
Comment: we changed the term allows to involves
- “Line 96 instead of “in turns” changes to “in turn”.”
Comment: we modified the typo.
- “Line 97-98 "evidence also suggests" should be "evidence also suggests".”
Comment: we modified the sentence as follow: “Recent evidence also suggests”
- “Line 112,118,119, 120 please italicize the gene name.”
Comment; we italicized gene names.
- “Line 146 Please write the full of “ASIA”.”
Comment; we apologize for the misunderstanding: Asia is continent, not an acronym. We modified in lowercase letter.
- “Line 157, please change “carotid o radial artery” to “carotid radial artery””
Comment: we modified the sentence as follow: “at carotid or at radial artery”
- “In Table 1 the author for AGE/RAGE has written See above. Please be specific or write some bullet points.”
Comment: we modified the Table accordingly to the Reviewer’s suggestion.

Reviewer 2 Report
Authors
Aging of arterial systems are associated to arterial stiffness in pathophysiologic concept in cardiovascular diseases. Increasing evidences support the view that “aging of arterial stiffness” serves also as indicator of the individual´s general health status or trigger to develop several pathologies as cardiovascular disease. Since the review refers to age in vascular systems, it should better focus the rationale of the review.
Aging vascular is a broad subject to be reviewed in depth as a single manuscript. In addition, several recent reviews have partially covered the topic of interest (doi: 10.3389/fcvm.2023.1037227. eCollection 2023; doi: 10.3390/jcdd10020044; doi: 10.1007/s40292-022-00555-0. Epub 2022 Dec 12; doi: 10.36660/abc.20210708; doi: 10.1038/s42003-022-03563-x). To this respect, to increase novelty, the authors should focus the manuscript in discussing the most recent findings on key concepts in the field of aging and arterial stiffness, referring the reader to recent comprehensive reviews covering a more general overview of the topic.
Several original studies relate to aging and vascular systems need to be discussed in detail in the manuscript. Indeed, the relationship aging and vascular systems are well-known. Therefore, a novel structure of the manuscript is required. The authors should consider the top-10 clinical and behavior risk factors for arterial stiffness by comparison between 10 years ago compared to today, according to pubmed frequency available online. Evidence of relationship with aging of arterial stiffness, molecular signaling and animal models should be included in the new version of manuscript to the two risk factors.
Other studies based on pharmacological treatment cannot clearly discern between a direct effect of drug on arterial stiffness or rather secondary effect to another diseases, according to the authors. Thus, the association between pharmacological treatment and arterial stiffness will be beyond the subject of the manuscript, except lipid-lowering drugs.
Minor comments
Please include Figures and bibliographic references from which the given information was extracted.
Tables with detailed content should be considered.
Author Response
Reply to Reviewer 2
- Aging of arterial systems are associated to arterial stiffness in pathophysiologic concept in cardiovascular diseases. Increasing evidences support the view that “aging of arterial stiffness” serves also as indicator of the individual´s general health status or trigger to develop several pathologies as cardiovascular disease. Since the review refers to age in vascular systems, it should better focus the rationale of the review.Aging vascular is a broad subject to be reviewed in depth as a single manuscript. In addition, several recent reviews have partially covered the topic of interest (doi: 3389/fcvm.2023.1037227. eCollection 2023; doi: 10.3390/jcdd10020044; doi: 10.1007/s40292-022-00555-0. Epub 2022 Dec 12; doi: 10.36660/abc.20210708; doi: 10.1038/s42003-022-03563-x). To this respect, to increase novelty, the authors should focus the manuscript in discussing the most recent findings on key concepts in the field of aging and arterial stiffness, referring the reader to recent comprehensive reviews covering a more general overview of the topic. Several original studies relate to aging and vascular systems need to be discussed in detail in the manuscript. Indeed, the relationship aging and vascular systems are well-known. Therefore, a novel structure of the manuscript is required. The authors should consider the top-10 clinical and behavior risk factors for arterial stiffness by comparison between 10 years ago compared to today, according to pubmed frequency available online. Evidence of relationship with aging of arterial stiffness, molecular signaling and animal models should be included in the new version of manuscript to the two risk factors.”
Comment: We understand the point of view of the Reviewer because age in vascular system is a wide topic which cannot be cover by a single paper. However, we wish to point out that the aim of our work was to present a specific theme which narrowed the above-mentioned wide topic. Indeed, we decided to focus on the aging of arterial stiffness, and we think that the structure of the review is consistent with the aim of our work. Therefore, we appreciated the Reviewer’s suggestion to organize our manuscript by comparing top-10 clinical and behavior risk factor for arterial stiffness across 10 years. But at the same time, we think that Reviewer’s remark deserves an additional and different article, which is far from our aim.
- “Other studies based on pharmacological treatment cannot clearly discern between a direct effect of drug on arterial stiffness or rather secondary effect to another diseases, according to the authors. Thus, the association between pharmacological treatment and arterial stiffness will be beyond the subject of the manuscript, except lipid-lowering drugs.”
Comment: We partially agree with Reviewer’s comment. Indeed, effects of drugs on arterial stiffness reported in previous derived by the treatment of other specific diseases. As stated in the manuscript, no drug has been synthetized to specifically treat the arterial stiffness. Therefore, if one decides to exclude antihypertensive and anti-diabetic drugs, to be more consistent we should also exclude lipid lowering drug. At the same time, we think that paragraph entitled “Pharmacological treatment of arterial stiffness” is consistent with the structure of our study and we wish to include in our manuscript. We would ask the Editor to choose if remove or let this paragraph.
- “Please include Figures and bibliographic references from which the given information was extracted.Tables with detailed content should be considered”.
Comment: We agree with Reviewer’s comment. References were added to the Table 2, which has also been detailed.

Reviewer 3 Report
This is a very nice review regarding aging and arterial stiffness. The only suggestion I would have is that the authors should consider a schematic in the paper which summarizes the items being discussed.
Author Response
Reply to Reviewer 3
- “This is a very nice review regarding aging and arterial stiffness. The only suggestion I would have is that the authors should consider a schematic in the paper which summarizes the items being discussed.”
Comment: we wish to thank the Reviewer for appreciating our study. We summarized the items discussed in our review on Figure 1, included in the revised version of our manuscript.

Round 2
Reviewer 2 Report
The response given by the authors is understood by the reviewer. However, The paper needs novelty as contribution to a known topic.
Author Response
- “The response given by the authors is understood by the reviewer. However, The paper needs novelty as contribution to a known topic.”
Comment: we added a paragraph on the effect of the novel drugs gliflozins on arterial stiffness.
Round 3
Reviewer 2 Report
The authors should add population and cohort studies in the different sections of the manuscript (collagen and elastin, calcification, .........)
Author Response
- “The authors should add population and cohort studies in the different sections of the manuscript (collagen and elastin, calcification, .........).”
Comment: we wish to thank the Reviewer for her/his suggestion, We added population and cohort studies in all appropriate section of the manuscript..
